# Learning the Local Statistics of Optical Flow

**Dan Rosenbaum**[1], **Daniel Zoran**[2], **Yair Weiss**[1,2]
[1] CSE , [2] ELSC , Hebrew University of Jerusalem
{danrsm,daniez,yweiss}@cs.huji.ac.il

## Abstract

Motivated by recent progress in natural image statistics, we use newly available datasets with ground truth optical flow to learn the local statistics of optical flow and compare the learned models to prior models assumed by computer vision researchers. We find that a Gaussian mixture model (GMM) with 64 components provides a significantly better model for local flow statistics when compared to commonly used models. We investigate the source of the GMM's success and show it is related to an explicit representation of flow boundaries. We also learn a model that jointly models the local intensity pattern and the local optical flow. In accordance with the assumptions often made in computer vision, the model learns that flow boundaries are more likely at intensity boundaries. However, when evaluated on a large dataset, this dependency is very weak and the benefit of conditioning flow estimation on the local intensity pattern is marginal.

## 1 Introduction

Sintel MPI                    KITTI

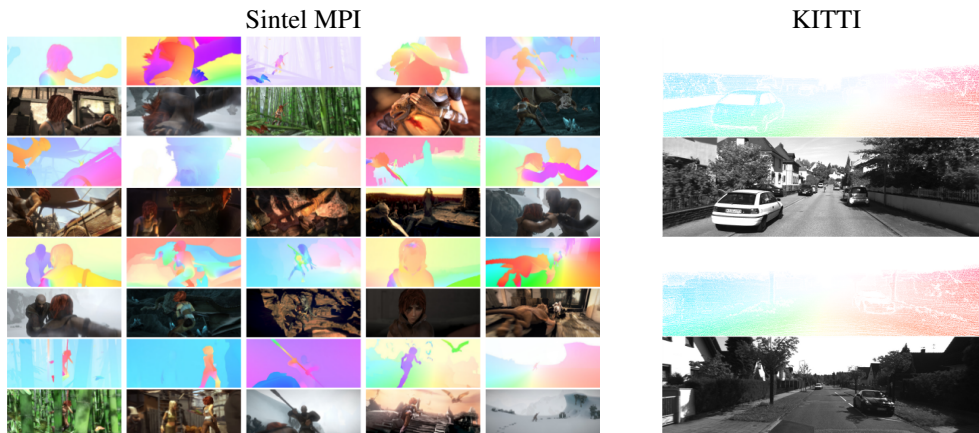

Figure 1: Samples of frames and flows from new flow databases. We leverage these newly available resources to learn the statistics of optical flow and compare this to assumptions used by computer vision researchers.

The study of natural image statistics is a longstanding research topic with both scientific and engineering interest. Recent progress in this field has been achieved by approaches that systematically compare different models of natural images with respect to numerical criteria such as log likelihood on held-out data or coding efficiency [1, 10, 14]. Interestingly, the best models in terms of log likelihood, when used as priors in image restoration tasks, also yield state-of-the-art performance [14].

Many problems in computer vision require good priors. A notable example is the computation of *optical flow*: a vector at every pixel that corresponds to the two dimensional projection of the motion

at that pixel. Since local motion information is often ambiguous, nearly all optical flow estimation algorithms work by minimizing a cost function that has two terms: a local data term and a "prior" term (see. e.g. [13, 11] for some recent reviews).

Given the success in image restoration tasks, where learned priors give state-of-the-art performance, one might expect a similar story in optical flow estimation. However, with the notable exception of [9] (which served as a motivating example for this work and is discussed below) there have been very few attempts to learn priors for optical flow by modeling local statistics. Instead, the state-of-the-art methods still use priors that were formulated by computer vision researchers. In fact, two of the top performing methods in modern optical flow benchmarks use a hand-defined smoothness constraint that was suggested over 20 years ago [6, 2].

One big difference between image statistics and flow statistics is the availability of ground truth data. Whereas for modeling image statistics one merely needs a collection of photographs (so that the amount of data is essentially unlimited these days), for modeling flow statistics one needs to obtain the ground truth motion of the points in the scene. In the past, the lack of availability of ground truth data did not allow for learning an optical flow prior from examples. In the last two years, however, two ground truth datasets have become available. The Sintel dataset (figure 1) consists of a thousand pairs of frames from a highly realistic computer graphics film with a wide variety of locations and motion types. Although it is synthetic, the work in [3] convincingly show that both in terms of image statistics and in terms of flow statistics, the synthetic frames are highly similar to real scenes. The KITTI dataset (figure 1) consists of frames taken from a vehicle driving in a European city [5]. The vehicle was equipped with accurate range finders as well as accurate localization of its own motion, and the combination of these two sources allow computing optical flow for points that are stationary in the world. Although this is real data, it is sparse (only about 50% of the pixels have ground truth flow).

In this paper we leverage the availability of ground truth datasets to learn explicit statistical models of optical flow. We compare our learned model to the assumptions made by computer vision algorithms for estimating flow. We find that a Gaussian mixture model with 64 components provides a significantly better model for local flow statistics when compared to commonly used models. We investigate the source of the GMM's success and show that it is related to an explicit representation of flow boundaries. We also learn a model that jointly models the local intensity pattern and the local optical flow. In accordance with the assumptions often made in computer vision, the model learns that flow boundaries are more likely at intensity boundaries. However, when evaluated on a large dataset, this dependency is very weak and the benefit of conditioning flow estimation on the local intensity pattern is marginal.

### 1.1 Priors for optical flow

One of the earliest methods for optical flow that is still used in applications is the celebrated Lucas-Kanade algorithm [7]. It overcomes the local ambiguity of motion analysis by assuming that the optical flow is constant within a small image patch and finds this constant motion by least-squares estimation. Another early algorithm that is still widely used is the method of Horn and Schunck [6]. It finds the optical flow by minimizing a cost function that has a data term and a "smoothness" term. Denoting by $u$ the horizontal flow and $v$ the vertical flow, the smoothness term is of the form:

$$J_{HS} = \sum_{x,y} u_x^2 + u_y^2 + v_x^2 + v_y^2$$

where $u_x, u_y$ are the spatial derivatives of the horizontal flow $u$ and $v_x, v_y$ are the spatial derivatives of the vertical flow $v$. When combined with modern optimization methods, this algorithm is often among the top performing methods on modern benchmarks [11, 5].

Rather than using a quadratic smoothness term, many authors have advocated using more robust terms that would be less sensitive to outliers in smoothness. Thus the Black and Anandan [2] algorithm uses:

$$J_{BA} = \sum_{x,y} \rho(u_x) + \rho(u_y) + \rho(v_x) + \rho(v_y)$$

where $\rho(t)$ is a function that grows slower than a quadratic. Popular choices for $\rho$ include the Lorentzian, the truncated quadratic and the absolute value $\rho(x) = |x|$ (or a differentiable approximation to it $\rho(x) = \sqrt{\epsilon + x^2}$)[11]. Both the Lorentzian and the absolute value robust smoothness

terms were shown to outperform quadratic smoothness in [11] and the absolute value was better among the two robust terms.

Several authors have also suggested that the smoothness term be based on the local intensity pattern, since motion discontinuities are more likely to occur at intensity boundaries. Ren [8] modified the weights in the Lucas and Kanade least-squares estimation so that pixels that are on different sides of an intensity boundary will get lower weights. In the context of Horn and Shunck, several authors suggest using weights to the horizontal and vertical flow derivatives, where the weights had an inverse relationship with the image derivatives: large image derivatives lead to low weight in the flow smoothness (see [13] and references within for different variations on this idea). Perhaps the simplest such regularizer is of the form:

$$J_{HSI} = \sum_{x,y} w(I_x)(u_x^2 + v_x^2) + w(I_y)(u_y^2 + v_y^2) \tag{1}$$

As we discuss below, this prior can be seen as a Gaussian prior on the flow that is *conditioned on the intensity*.

In contrast to all the previously discussed priors, Roth and Black [9] suggested learning a prior from a dataset. They used a training set of optical flow obtained by simulating the motion of a camera in natural range images. The prior learned by their system was similar to a robust smoothness prior, but the filters are not local derivatives but rather more random-looking high pass filters. They did not observe a significant improvement in performance when using these filters, and standard derivative filters are still used in most smoothness based methods.

Given the large number of suggested priors, a natural question to ask is: what is the best prior to use? One way to answer this question is to use these priors as a basis for an optical flow estimation algorithm and see which algorithm gives the best performance. Although such an approach is certainly informative it is difficult to get a definitive answer using it. For example, Sun et al. [11] reported that adding a non-local smoothness term to a robust smoothness prior significantly improved results on the Middlebury benchmark, while Geiger et al. [5] reported that this term decreased performance on KITTI benchmark. Perhaps the main difficulty with this approach is that the prior is only one part of an optical flow estimation algorithm. It is always combined with a non-convex likelihood term and optimized using a nonlinear optimization algorithm. Often the parameters of the optimization have a very large influence on the performance of the algorithm.

In this paper we take an alternative approach. Motivated by recent advances in natural image statistics and the availability of new datasets, we compare different priors in terms of (1) log likelihood on held-out data and (2) inference performance with tractable posteriors. Our results allow us to rigorously compare different prior assumptions.

## 2 Comparing priors as density models

In order to compare different prior models as density models, we generate a training set and test set of optical flow patches from the ground truth databases. Denoting by $f$ a single vector that concatenates all the optical flow in a patch (e.g. if we consider $8 \times 8$ patches, $f$ is a vector of length 128 where the first 64 components denote $u$ and the last 64 components denote $v$). Given a prior probability model $\Pr(f; \theta)$ we use the training set to estimate the free parameters of the model $\theta$ and then we measure the log likelihood of *held out* patches from the test set.

From Sintel, we divided the pairs of frames for which ground truth is available into 708 pairs which we used for training and 333 pairs which we used for testing. The data is divided into scenes and we made sure that different scenes are used in training and testing. We created a *second test set* from the KITTI dataset by choosing a subset of patches for which full ground truth flow was available. Since we only consider full patches, this set is smaller and hence we use it only for testing, not for training.

The priors we compared are:

- Lucas and Kanade. This algorithm is equivalent to the assumption that the observed flow is generated by a constant $(u_0, v_0)$ that is corrupted by IID Gaussian noise. If we also assume

that $u_0, v_0$ have a zero mean Gaussian distribution, $\Pr(f)$ is a zero mean multidimensional Gaussian with covariance given by $\sigma_p^2 OO^t + \sigma_n^2 I$ where $O$ is a binary $128 \times 2$ matrix and $\sigma_p$ the standard deviation of $u_0, v_0$ and $\sigma_n$ the standard deviation of the noise.

- Horn and Schunck. By exponentiating $J_{HS}$ we see that $\Pr(f; \theta)$ is a multidimensional Gaussian with covariance matrix $\lambda DD^T$ where $D$ is a $256 \times 128$ derivative matrix that computes the derivatives of the flow field at each pixel and $\lambda$ is the weight given to the prior relative to the data term. This covariance matrix is not positive definite, so we use $\lambda DD^T + \epsilon I$ and determine $\lambda, \epsilon$ using maximum likelihood.

- L1. We exponentiate $J_{BA}$ and obtain a multidimensional Laplace distribution. As in Horn and Schunck, this distribution is not normalizeable so we multiply it by an IID Laplacian prior on each component with variance $1/\epsilon$. This again gives two free parameters $(\lambda, \epsilon)$ which we find using maximum likelihood. Unlike the Gaussian case, the solution of the ML parameters and the normalization constant cannot be done in closed form, and we use Hamiltonian Annealed Importance Sampling [10].

- Gaussian Mixture Models (GMM). Motivated by the success of GMMs in modeling natural image statistics [14] we use the training set to estimate GMM priors for optical flow. Each mixture component is a multidimensional Gaussian with full covariance matrix and zero mean and we vary the number of components between 1 and 64. We train the GMM using the standard Expectation-Maximization (EM) algorithm using mini-batches. Even with a few mixture components, the GMM has far more free parameters than the previous models but note that we are measuring success on *held out* patches so that models that overfit should be penalized.

The summary of our results are shown in figure 2 where we show the mean log likelihood on the Sintel test set. One interesting thing that can be seen is that the local statistics validate some assumptions commonly used by computer vision researchers. For example, the Horn and Shunck smoothness prior is as good as the *optimal Gaussian prior* (GMM1) even though it uses local first derivatives. Also, the robust prior (L1) is much better than Horn and Schunck. However, as the number of Gaussians increase the GMM is significantly better than a robust prior on local derivatives.

A closer inspection of our results is shown in figure 3. Each figure shows the histogram of log likelihood of held out patches: the more shifted the histogram is to the right, the better the performance. It can be seen that the GMM is indeed much better than the other priors including cases where the test set is taken from KITTI (rather than Sintel) and when the patch size is $12 \times 12$ rather than $8 \times 8$.

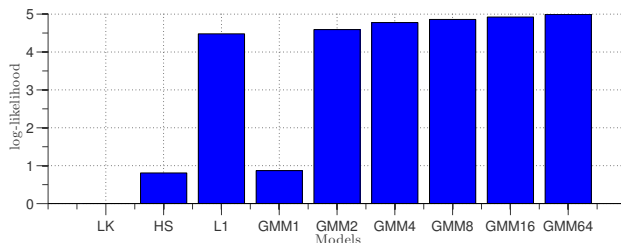

Figure 2: mean log likelihood of the different models for $8 \times 8$ patches extracted from held out data from Sintel. The GMM outperforms the models that are assumed by computer vision researchers.

## 2.1 Comparing models using tractable inference

A second way of comparing the models is by their ability to restore corrupted patches of optical flow. We are not claiming that optical flow restoration is a real-world application (although using priors to "fill in" holes in optical flow is quite common, e.g. [12, 8]). Rather, we use it because for the models we are discussing the inference can either be done in closed form or using convex optimization, so we would expect that better priors will lead to better performance.

We perform two flow restoration tasks. In "flow denoising" we take the ground truth flow and add IID Gaussian noise to all flow vectors. In "flow inpainting" we add a small amount of noise to all

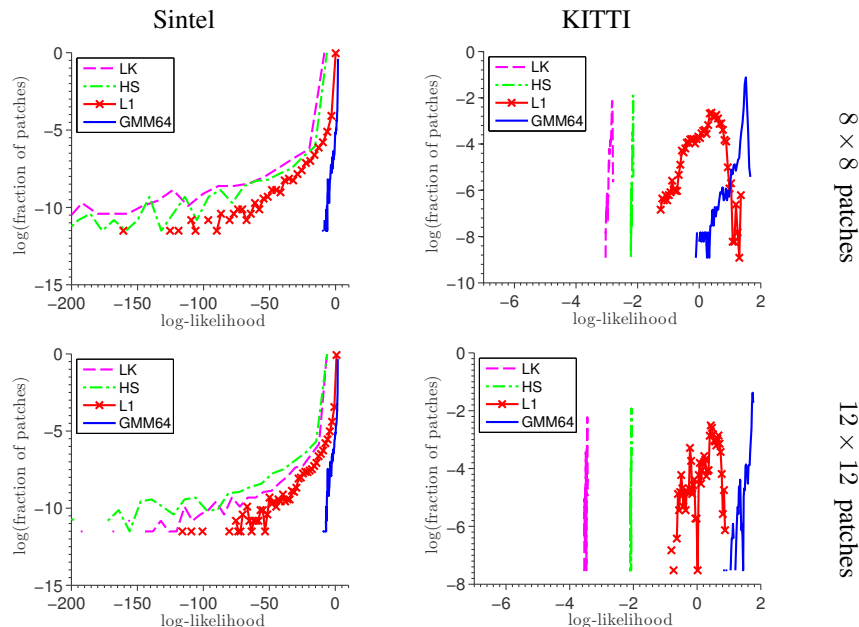

Figure 3: Histograms of log-likelihood of different models on the KITTI and Sintel test sets with two different patch sizes. As can be seen, the GMM outperforms other models in all four cases.

flow vectors and a very big amount of noise to some of the flow vectors (essentially meaning that these flow vectors are not observed). For the Gaussian models and the GMM models the Bayesian Least Squares (BLS) estimator of $f$ given $y$ can be computed in closed form. For the Laplacian model, we use MAP estimation which leads to a convex optimization problem. Since MAP may be suboptimal for this case, we optimize the parameters $\lambda, \epsilon$ for MAP inference performance.

Results are shown in figures 4,5. The standard deviation of the ground truth flow is approximately 11.6 pixels and we add noise with standard deviations $10, 20$ and $30$ pixel. Consistent with the log likelihood results, L1 outperforms the Gaussian methods but is outperformed by the GMM. For small noise values the difference between L1 and the GMM is small, but as the amount of noise increases L1 becomes similar in performance to the Gaussian methods and is much worse than the GMM.

## 3 The secret of the GMM

We now take a deeper look at how the GMM models optical flow patches. The first (and not surprising) thing we found is that the covariance matrices learned by the model are block diagonal (so that the $u$ and $v$ components are independent given the assignment to a particular component).

More insight can be gained by considering the GMM as a local subspace model: a patch which is generated by component $k$ is generated as a linear combination of the eigenvectors of the $k$th covariance. The coefficients of the linear combination have energy that decays with the eigenvalue: so each patch can be well approximated by the leading eigenvectors of the corresponding covariance. Unlike global subspace models, different subspace models can be used for different patches, and during inference with the model one can infer which local subspace is most likely to have generated the patch.

Figure 6 shows the dominant leading eigenvectors of all 32 covariance matrices in the GMM32 model: the eigenvectors of $u$ are followed by the eigenvectors of $v$. The number of eigenvectors displayed in each row is set so that they capture 99% of the variance in that component. The rows are organized by decreasing mixing weight. The right hand half of each row shows (u,v) patches that are sampled from that Gaussian.

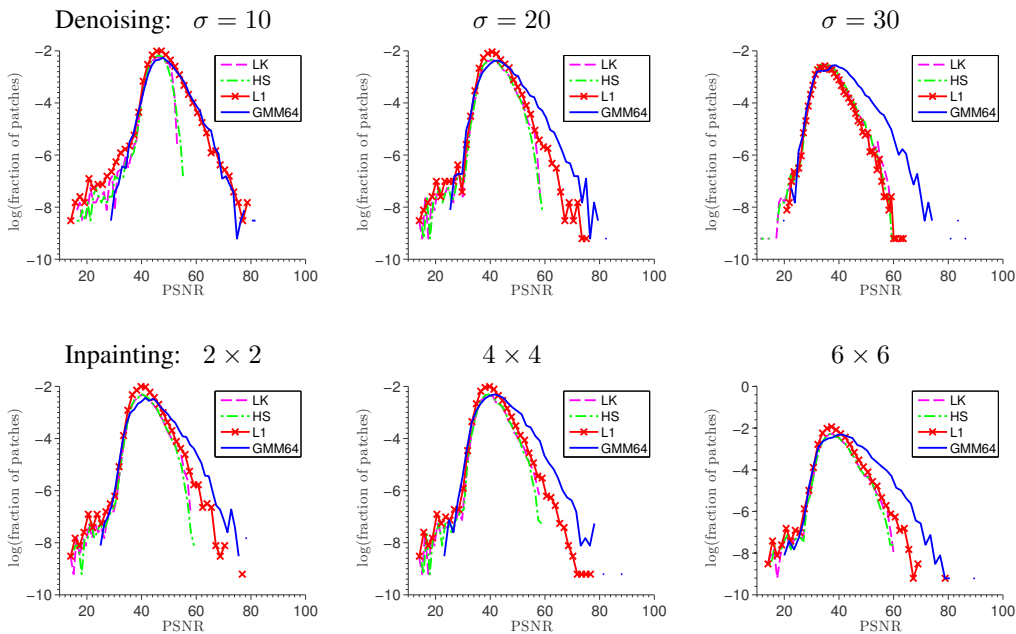

Figure 4: Denoising with different noise values and inpainting with different hole sizes.

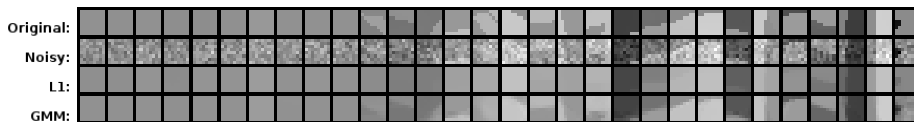

Figure 5: Visualizing denoising performance ($\sigma = 30$).

It can be seen that the first 10 components or so model very smooth components (in fact the samples appear to be completely flat). A closer examination of the eigenvalues shows that these ten components correspond to smooth motions of different speeds. This can also be seen by comparing the v samples on the top row which are close to gray with those in the next two rows which are much closer to black or white (since the models are zero mean, black and white are equally likely for any component).

As can be seen in the figure, almost all the energy in the first components is captured by uniform motions. Thus these components are very similar to a *non-local* smoothness assumption similar to the one suggested in [11]): they not only assume that derivatives are small but they assume that all the $8 \times 8$ patch is constant. However, unlike the suggestion in [11] to enforce non-local smoothness by applying a median filter at *all pixels*, the GMM only applies non-local smoothness at a subset of patches that are inferred to be generated by such components.

As we go down in the figure towards more rare components. we see that the components no longer model flat components but rather motion boundaries. This can be seen both in the samples (rightmost rows) and also in the leading eigenvectors (shown on the left) which each control one side of a boundary. For example, the bottom row of the figure illustrates a component that seems to generate primarily diagonal motion boundaries.

Interestingly, such local subspace models of optical flow have also been suggested by Fleet et al. [4]. They used synthetic models of moving occlusion boundaries and bars to learn linear subspace models of the flow. The GMM seems to support their intuition that learning separate linear subspace models for flat vs motion boundary is a good idea. However, unlike the work of Fleet et al. the separation into "flat" vs. "motion boundary" was learned in an unsupervised fashion directly from the data.

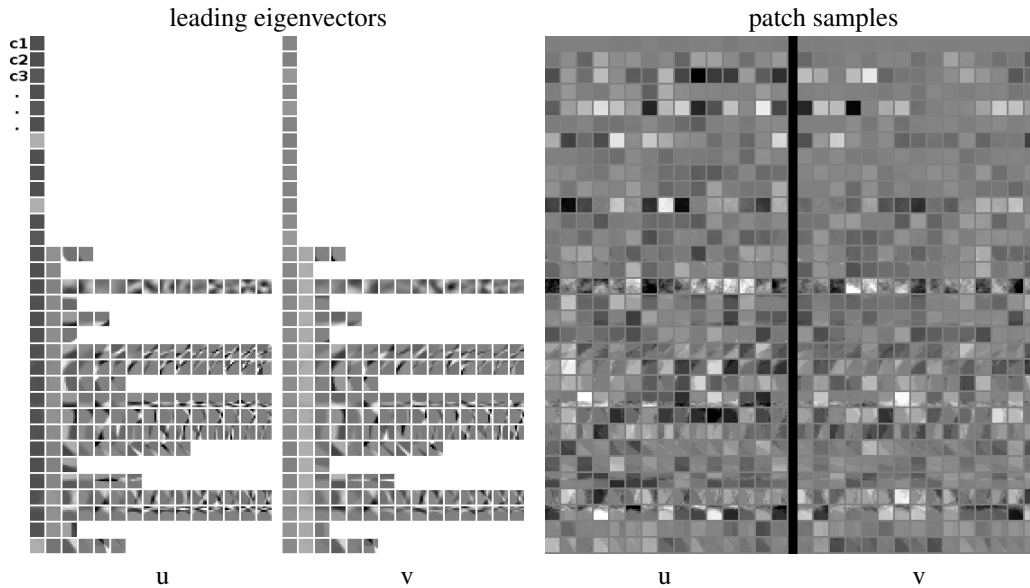

Figure 6: The eigenvectors and samples of the GMM components. GMM is better because it explicitly models edges and flat patches separately.

## 4 A joint model for optical flow and intensity

As mentioned in the introduction, many authors have suggested modifying the smoothness assumption by conditioning it on the local intensity pattern and giving a higher penalty for motion discontinuities in the absence of intensity discontinuities. We therefore ask, does conditioning on the local intensity give better log likelihood on held out flow patches? Does it give better performance in tractable inference tasks?

We evaluated two flow models that are conditioned on the local intensity pattern. The first one is a conditional Gaussian (eq. 1) with exponential weights, i.e. $w(I_x) = \exp(-I_x^2/\sigma^2)$ and the variance parameter $\sigma^2$ is optimized to maximize performance. The second one is a Gaussian mixture model that simultaneously models both intensity and flow.

The simultaneous GMM we use includes a 200 component GMM to model the intensity together with a 64 dimensional GMM to model the flow. We allow a dependence between the hidden variable of the intensity GMM and that of the flow GMM. This is equivalent to a hidden Markov model (HMM) with 2 hidden variables: one represents the intensity component and one represents the flow component (figure 8). We learn the HMM using the EM algorithm. Initialization is given by independent GMMs learned for the intensity (we actually use the one learned by [14] which is available on their website) and for the flow. The intensity GMM is not changed during the learning. Conditioned on the intensity pattern, the flow distribution is still a GMM with 64 components (as in the previous section) but the mixing weights depend on the intensity.

Given these two conditional models, we now ask: will the conditional models give better performance than the unconditional ones? The answer, shown in figure 7 was surprising (to us). Conditioning on the intensity gives basically zero improvement in log likelihood and a slight improvement in flow denoising only for very large amounts of noise. Note that for all models shown in this figure, the denoised estimate is the Bayesian Least Squares (BLS) estimate, and is optimal given the learned models.

To investigate this effect, we examine the transition matrix between the intensity components and the flow components (figure 8). If intensity and flow were independent, we would expect all rows of the transition matrix to be the same. If an intensity boundary always lead to a flow boundary, we would expect the bottom rows of the matrix to have only one nonzero element. By examining the learned transition matrix we find that while there is a dependency structure, it is not very strong.

Regardless of whether the intensity component corresponds to a boundary or not, the most likely flow components are flat. When there is an intensity boundary, the flow boundary in the same orientation becomes more likely. However, even though it is more likely than in the unconditioned case, it is still less likely than the flat components.

To rule out that this effect is due to a local optimum found by EM, we conducted additional experiments whereby the emission probabilities were held fixed to the GMMs learned independently for flow and motion and each patch in the training set was assigned one intensity and one flow component. We then estimated the joint distribution over flow and motion components by simply counting the relative frequency in the training set. The results were nearly identical to those found by EM.

In summary, while our learned model supports the standard intuition that motion boundaries are more likely at intensity boundaries, it suggests that when dealing with a large dataset with high variability, there is very little benefit (if any) in conditioning flow models on the local intensity.

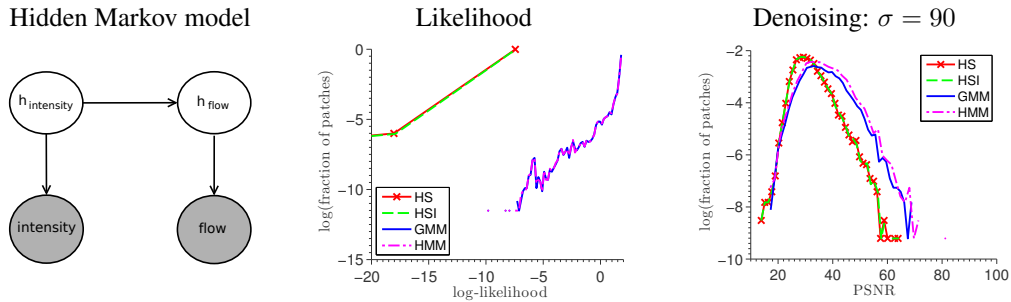

Figure 7: The hidden Markov model we use to jointly model intensity and flow. Both log likelihood and inference evaluations show almost no improvement of conditioning flow on intensity.

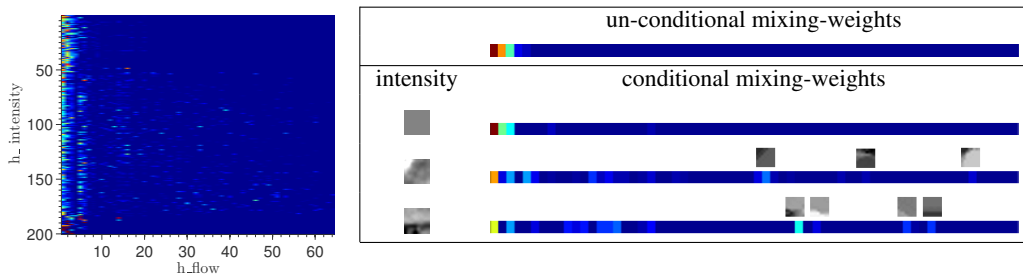

Figure 8: Left: the transition matrix learned by the HMM. Right: comparing rows of the matrix to the unconditional mixing weights. Conditioned on an intensity boundary, motion boundaries become more likely but are still less likely than a flat motion.

## 5 Discussion

Optical flow has been an active area of research for over 30 years in computer vision, with many methods based on assumed priors over flow fields. In this paper, we have leveraged the availability of large ground truth databases to learn priors from data and compare our learned models to the assumptions typically made by computer vision researchers. We find that many of the assumptions are actually supported by the statistics (e.g. the Horn and Schunck model is close to the optimal Gaussian model, robust models are better, intensity discontinuities make motion discontinuities more likely). However, a learned GMM model with 64 components significantly outperforms the standard models used in computer vision, primarily because it explicitly distinguishes between flat patches and boundary patches and then uses a different form of nonlocal smoothness for the different cases.

#### Acknowledgments

Supported by the Israeli Science Foundation, Intel ICRI-CI and the Gatsby Foundation.

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
