[Reviews · NeurIPS 2013]

Submitted by Assigned_Reviewer_1

This paper attempts to learn a model of optic flow based on a Gaussian Mixture Model (GMM). Following previous successful application of GMM's to the intensity distributions in natural images, the authors here train a GMM to the optic flow in image sequences and show that it results in a better model (in terms of log-likelihood) of optic flow patterns. The GMM's for intensity and optic flow are combined using a hidden markov model in which the intensity model conditions the optic flow model. Application to denoising and filling are demonstrated.

The only things that aroused my concern were the relatively small size of the training set and the way they trained their joint intensity/flow model. ~700 training examples struck me as a bit low for a GMM with 128 dimensions and >10 mixture components.

Learning the interaction between intensity and optic flow is nice, but what what about depth? That would seem to make more sense.

Minor comments:

Brackets in equation 1.

Figure 2, axis labels not legible.

"These are multiplied to form a 12, 800 component GMM .." - confusing

Figure 7: plot labels illegible
Summary: This is a natural extension of previous work applying GMM's to natural images. The application to optic flow makes sense and the results are good.

Submitted by Assigned_Reviewer_5

The paper proposes using recent large-scale optical flow datasets to learn priors on optical flow. Two types of priors are learned: a prior on flow vectors, and a joint prior on intensity and flow. GMMs are used for both priors. It is shown that these rich data-driven priors outperform previous ad-hoc priors on flow restoration tasks. On the other hand, the advantages of these priors for actual optical flow estimation are not demonstrated.

Overall, this is a nice paper, albeit a bit narrow. I am positive on it and support acceptance, in part because I work in the area and find some of the observations made in this paper to be useful.

Pros:

- Natural idea, it's good that somebody did it, nice to know what the results are.
- It's nice that the likelihood estimation results translate across datasets.
- The observation that flow boundaries are not strongly correlated with intensity boundaries is very nice and is the highlight of the paper. It's not completely clear that the evidence in the paper actually supports making this claim broadly and strongly, but the evidence is certainly suggestive. This observation is a stimulating addition to the literature. I would like to see it published.

Cons:

- Benefits to actual optical flow estimation are not demonstrated, and are potentially marginal. The paper's contribution seems to be mostly conceptual rather than practical.
- With regards to the intriguing results described in line 373 and onwards: could this be due to the suboptimal learning algorithm? Can we really draw general conclusions from these experiments, or should the conclusions be qualified a bit due to the limitations of the learning procedure? Or the data?

Specific/minor comments:

- lines 73, 74: "(figure 1b)", "city [5]"
- line 82: "GMM's", "show that it is"
- equation (1): fix parentheses and brackets in multiple places
- line 355: "of of" -> "of"
- lines 300-323 and figure 6: I found it hard to follow the text and to see precisely the phenomena that are being described in Figure 6. The figure should be annotated much better. The phenomena should be visually apparent. I just don't see some of the things that are being described in the figure.
- lines 357-358: what intensity weighting did you use for this model and why?
- line 424: "assumed priors"

Summary: Nice paper, a bit academic, but will be of interest to the optical flow community. Accept.

Submitted by Assigned_Reviewer_6

The authors investigate different priors for optical flow, based on training data from the Sintel and KITTI datasets. Most of the studies are performed by expressing the priors as density models and measuring performance in terms of log-likelihood metrics, but results for flow in-painting and denoising are also given. The authors find good agreement with the performance of earlier robust smoothness priors but find a trained GMM with 64 components to be superior, under the studied metrics.

This paper is clearly presented, with good motivation, experiments and insights. It is more of an empirical evaluation for optical flow paper than a methodology paper, which makes it somewhat less appealing for NIPS. Even so, what fits the conference may be a matter of taste and I wouldn’t make a strong judgment here. My two concerns are the following: (1) most of the studies are using the log-likelihood metric, thus glossing over the important problem of model complexity. Wouldn't it be expected that more complex models perform better, in terms of density modeling anyway? Analyzing just log-likelihoods even on test data does not seem entirely conclusive.(2) Although useful experiments are performed in isolation, focusing on density models or on somewhat simper tasks like inpainting or denoising, there is no evidence that the winning 64 component GMM would be superior in a complete optical flow estimation pipeline. This would be the most relevant test to perform, isn’t it?
Summary: An interesting study on the impact of trained regularizers in modeling optical flow data. Clear presentation and relevant quantitative studies, perhaps skewed towards not entirely relevant log-likelihood metrics and without assessing the impact of the winning model in a full optical flow estimation pipeline.
Author Feedback

Author rebuttal: We thank the reviewers for their helpful comments and we are happy they like the paper.

In response to reviewer 6’s concern about model complexity vs. likelihood performance:
Indeed complex models are expected to have higher likelihood on the TRAINED data. The result we demonstrate in figures 2 and 3 are of the likelihood on HELD-OUT data which was not used for training. This serves to demonstrate the training method and the data-set that were used do not lead to overfitting. In figure 3 we also show the same hold for a completely different data-set (KITTI) which suggests the models learn a global phenomena that is not specific to the Sintel data-set.

In response to reviewers 5 and 6’s concern about the absence of a complete optical flow estimation pipeline:
We agree that developing a complete optical flow estimation pipeline which achieves higher accuracy than current methods is of high interest. Nevertheless, such a pipeline combines the choice of different prior models, different noise models and different optimization methods. This makes it hard to understand the reason one estimation method performs better than the other (see reference 11 in the paper for such an attempt). Therefore, we think that focusing on one aspect, namely the prior model, and rigorously comparing it to other models is of interest to the optical flow community and can serve to develop better estimation procedures.

In response to reviewer 5’s concern about the conclusions from the intensity/flow joint model experiments:
We agree that one should use caution in drawing general conclusions from these experiments and will emphasize this in the next version. We will also add more experiments that we have performed with different training procedures. The results are quite robust to different training procedures and consistently show that even with a joint model that learns a reasonable correlation between flow boundaries and intensity edges (figure 8), the improvement over a flow-only model is marginal.

In response to reviewer 1’s concern about the data-set size:
It is true that a data-set of 700 images is rather small but since we train models of optical flow patches rather than images, the effective size of the data-set is much larger (containing about 300 million patches). Figures 2 and 3 show that there is no overfitting in the training even with respect to different data-sets such as KITTI.